# Cation Disorder and Local Structural Distortions in Ag_x_Bi_1–x_S_2_ Nanoparticles

**DOI:** 10.3390/nano10020316

**Published:** 2020-02-12

**Authors:** Jagadesh Kopula Kesavan, Francesco d’Acapito, Paolo Scardi, Alexandros Stavrinadis, Mehmet Zafer Akgul, Ignasi Burgués-Ceballos, Gerasimos Konstantatos, Federico Boscherini

**Affiliations:** 1Department of Physics and Astronomy, University of Bologna, Viale C. Berti Pichat 6/2, 40127 Bologna, Italy; jagadesh.kopula@unibo.it; 2Istituto Officina dei Materiali, Consiglio Nazionale delle Ricerche, OGG, c/o ESRF, 71, Avenue des Martyrs, CS40220, CEDEX 9, 38043 Grenoble, France; dacapito@esrf.fr; 3Department of Civil, Environmental & Mechanical Engineering, University of Trento, Via Mesiano 77, 38123 Trento, Italy; paolo.scardi@unitn.it; 4ICFO-Institut de Ciencies Fotòniques, The Barcelona Institute of Science and Technology, 08860 Castelldefels (Barcelona), Spain; a.stavrinadis@gmail.com (A.S.); zafer.akgul@icfo.eu (M.Z.A.); ignasi.burgues@icfo.eu (I.B.-C.); gerasimos.konstantatos@icfo.eu (G.K.); 5ICREA—Institució Catalana de Recerca i Estudis Avançats, Passeig Lluís Companys 23, 08010 Barcelona, Spain

**Keywords:** nanoparticles, photovoltaics, X-ray absorption fine structure, X-ray diffraction, density functional simulations, molecular dynamics simulations

## Abstract

By combining X-ray absorption fine structure and X-ray diffraction measurements with density functional and molecular dynamics simulations, we study the structure of a set of Ag_x_Bi_1−x_S_2_ nanoparticles, a materials system of considerable current interest for photovoltaics. An apparent contradiction between the evidence provided by X-ray absorption and diffraction measurements is solved by means of the simulations. We find that disorder in the cation sublattice induces strong local distortions, leading to the appearance of short Ag–S bonds, the overall lattice symmetry remaining close to hexagonal.

## 1. Introduction

Dimetal chalcogenides of general formula I−V−VI_2_, in which I = Cu, Ag, or an alkali metal, V = Sb or Bi, and VI = S, Se, or Te, have recently attracted attention as an interesting class of materials for energy conversion from light (photovoltaics) and heat (thermoelectrics). Solar cells employing AgBiS_2_ nanoparticle (NP) photo-absorbing films of just 40 nm in thickness exhibit 6.4% certified photovoltaic efficiencies [1]. This unique feature is attributed to the material’s high optical extinction coefficient (>10^5^ cm^−1^), and to a nearly 100% internal quantum efficiency across the VIS and NIR parts of the optical spectrum; among other attractive features of the material is its environmental friendliness. The photovoltaic performance of AgBiS_2_ NPs is currently limited by the presence of electronic midgap trap states, which limit the maximum allowable thickness of films within a solar cell. The structural origin of these traps remains unknown and could be related to how Ag and Bi cations are distributed within the cation sub-lattice and/or to defects associated with structural distortions. Both AgBiS_2_ and AgBiSe_2_ have been demonstrated to have attractive thermoelectric properties due to the very low lattice thermal conductivity, a property related to bond anharmonicity [2,3,4].

The low-temperature phase of AgBiS_2_ is that of the mineral matildite (hexagonal lattice, space group 164, P$\bar{3}$*m*1), while above 179 °C, the stable phase is that of schapbachite (cubic lattice, space group 225, i.e., F*m*$\bar{3}$*m*) [5]. The two structures are related and differ in the relative arrangement of cations [6]. The matildite structure can be described in terms of a large cubic unit cell within which a smaller hexagonal one is found; the S atoms form a face-centered cubic (FCC) arrangement and Ag and Bi atoms are alternatively placed along the cubic (001) directions, forming a rocksalt structure. In schapbachite, the S atoms are also found in an FCC arrangement, but layers of BiS and AgS alternate in the (001) directions. Using density functional theory (DFT) simulations, Viñes et al. [6] predicted that the matildite structure is more stable than the schapbachite one and that the former is a semiconductor while the latter is a metal; the high photoactivity of the matildite structure has been attributed to different effective masses of electrons in the conduction band and holes in the valence band and a high optical absorption coefficient.

The relation between physical properties and the ordering of cations in dimetal chalcogenides has been an important theme of current research. An order–disorder phase transition in cubic AgBiS_2_ nanocrystals, proposed to explain calorimetric measurements, has been suggested to be responsible for the high thermoelectric efficiency [2], but no direct structural proof was provided. The gradual transition from matildite to schapbachite by means of cation rearrangement has been simulated by means of DFT [7]. Cation rearrangement from matildite to schapbachite was found to induce a gradual decrease in the band gap as a result of conduction band states shifting downward toward the valence band; intermediate cases of ordering were suggested to result in significant local structural disruption, but this issue was never addressed experimentally. Additionally, small perturbations in the Ag to Bi ratio (that is, in the stoichiometry) originating from variations in the synthesis conditions may introduce further local distortions. A similar phase transition was also observed in the selenide analog of AgBiS_2_, namely, AgBiSe_2_, in which the cation arrangement related to a hexagonal–rhombohedral–cubic phase transition was associated with switching between different types of conductivity (*p* or *n*) and that a high thermoelectric performance was obtained for a disordered cation arrangement [3]. For a disordered arrangement in the cation sublattice, it can be expected to result in a similar effect on the properties of AgBiS_2_. Thus, to have an atomistic understanding of this material’s properties and to guide the experimental efforts for improving the performance of AgBiS_2_-based technologies, it is crucial to explore the effects of cation disorder and stoichiometric perturbations in the Ag to Bi ratio.

In this paper, we address the relation between cation ordering and local structural distortions in a set of Ag_x_Bi_1−x_S_2_ NP thin films with purposely modified compositions, slightly below and above the stoichiometric 1:1 Ag to Bi ratio. We have used X-ray absorption fine structure (XAFS) and X-ray diffraction (XRD) aided by DFT ab initio and molecular dynamics (MD) simulations. Owing to its chemical selectivity and sensitivity, XAFS [8] is the premier experimental tool to measure interatomic distances in the first few coordination shells and to probe the relative atomic arrangement in condensed matter, in particular, in alloys; a further attractive feature is the applicability to ordered or disordered materials both in the bulk or as nanostructures. A limitation of XAFS is its insensitivity to lattice symmetry, which, along with the local atomic arrangement, also plays a fundamental role in determining physical properties. XRD is a standard characterization tool in this regard and its applicability to nanostructures has been recently greatly expanded, showing how refined information on symmetry, lattice strain, particle size, and shape can be obtained [9,10,11]. In this paper, the interpretation of the experimental data is greatly aided by DFT [12] and MD simulations, the efficiency and availability of which have constantly improved in recent years. We show that in Ag_x_Bi_1−x_S_2_ NPs, disorder in the cation distribution leads to significant local structural distortions. An apparent contradiction between XAFS and XRD is solved using DFT and MD simulations.

## 2. Materials and Methods

Colloidal Ag_x_Bi_1−x_S_2_ NP thin films were synthesized by a hot-injection low-temperature approach [1]. In this method, 0.8 mmol of silver acetate and 1 mmol of bismuth acetate are reacted with oleic acid; then, 1 mmol of bis (trimethylsilyl) sulfide (TMS) diluted with octadecene is injected into silver-bismuth oleate solution at 100 °C to form the AgBiS_2_ nanocrystals. To synthesize Ag_x_Bi_1−x_S_2_ NPs with four different x values, we adjusted the amount of silver precursor introduced into the reaction flask, keeping the other parameters of the reaction constant. Energy-dispersive X-ray (EDX) spectra acquisition and analysis were performed with a Helios Nanolab 600 (FEI Company) microscope combined with an X-Max detector and INCA^®^ system (Oxford Instruments). High-resolution transmission electron microscopy (TEM) measurements were performed at the Scientific and Technological Centers of the University of Barcelona (CCiT-UB). TEM micrographs were obtained using a JEOL 2100 microscope operating at an accelerating voltage of 200 kV. The samples were prepared by drop-casting a toluene-based colloidal NP solution on a holey carbon-coated grid. TEM (Figure 1) indicated an average size of 3.5–4 nm; the particles tended to show oval projections, not perfectly rounded, with a short axis of about 3 nm and a long axis of 4–5 nm; the thickness for all samples was ~100 nm. The correspondence between sample code and Ag concentration is summarized in Table 1, in which we report the reaction precursor ratio and values measured by EDX.

XAFS measurements at the Ag K (25.514 keV) and Bi L_III_ (13.409 keV) edges were performed at the BM08 (LISA) beamline of the European Synchrotron Radiation Facility (ESRF), Grenoble, France [13]. All measurements were carried out using a dynamically sagittal-focusing Si (311) monochromator [14], and the spectra were recorded in the fluorescence yield mode using a 12 element hyper-pure Ge detector [15]. Ag_2_S and Bi_2_S_3_ polycrystalline powders were measured in the transmission mode as references. The structure of Ag_2_S is that of the mineral acanthite (monoclinic lattice, space group 14, i.e., P2_1_/c) with two inequivalent Ag sites present, see Figure 2, left panel. XAFS data analysis in the extended energy range (EXAFS) was performed using the DEMETER package [16] and theoretical signals based on FEFF [17]. Simulations of the X-ray absorption near-edge structure (XANES) part of the XAFS spectra were performed using FDMNES [18].

X-ray diffraction measurements were performed using a PANalytical X’Pert PRO MPD Alpha1 powder diffractometer with Cu K_α_ radiation (λ = 1.5406 Å, 45 kV–40 mA) in the Scientific and Technological Centers of the University of Barcelona (CCiT-UB). The samples were prepared by drop-casting a concentrated dispersion of Ag_x_Bi_1−x_S_2_ NPs onto a clean glass slide. The XRD patterns were analyzed with the TOPAS software for structural refinement [19], version 7, using the functions recently developed for the analysis of nanocrystalline powders [20]. Different structures were implemented in the refinement to assess the degree of matching to the experimental data. Structural information on standard phases was taken from the “materials project” web interface [21]. In addition, to fit quality and structural information, data analysis provides the crystalline domain size, assuming spherical shape with a lognormal distribution of diameters, and inhomogeneous strain (also referred to as microstrain). Details on the procedure are reported elsewhere [20,22].

Tentative structures for the clusters were simulated using the DFT method, as implemented in the VASP code (version 5.2) [12]. The structures considered were the following: Bi-substituted acanthite (AcB), schapbachite (Sch), matildite (Mat), and matildite with random cation site occupation (MaR). Pure acanthite (Aca), Bi_2_S_3_ (BiS), and sulfur (S) were also studied for the determination of the formation energies. The AcB cell was derived from that of Aca by substituting a Ag-II^+^ ion with a Bi^3+^ one and eliminating two Ag^+^ ions, which were located as far as possible from Bi^3+^ and from each other to ensure charge balance. A calculation carried out starting from Ag-I^+^ led to a similar local structure and total energy, so these two structures will be considered as equivalent. For this compound, the first structural relaxation was carried out on a small cell (10 atoms) followed by a final relaxation on an 80 atom cell (subsequently also used for the MD run). Schapbachite was obtained by filling the cation sites randomly with Ag and Bi. A similar process was applied for random matildite: The cation sites were considered all equal and randomly filled with either Bi or Ag. Table 2 summarizes the main details of the cells used for the structural simulation.

For each case, the geometry of the energy-minimized (relaxed) structure was determined, as well as the formation energy per atom and the simulated XAFS spectrum. For the structural relaxation, the PBE exchange-correlation functional [27] was used with a plane wave cut-off energy of 650 eV. For the cell energy calculation, increasingly dense k meshes were used until the energy difference between consecutive steps was less than 1/1000 of the total cell energy. The k point meshes were all centered on the Γ point and ranged from k = {3 × 3 × 3} for sulphur to k = {5 × 5 × 5} for the other compounds and k = {8 × 8 × 4} for hexagonal matildite. A Gaussian distribution with a smearing of 0.05 eV was used to populate the electronic states. The electronic density was considered converged when the difference between successive steps was less than 1 μeV. The structural relaxation process was considered converged when Feynman−Hellman forces were below 2 × 10^−4^ eV/Å; every 20 ionic steps, the calculation was stopped and restarted with a new plane wave base recalculated with the new structure.

Formation energies of the cells (and related energies per atom) were calculated starting from chemical potentials of each element $\mu_{\alpha }$ (α = Ag, Bi, S). These potentials were obtained from the cell energies E_y_ of acanthite, Bi_2_S_3_, and S, and the number of each species in the cell N_w_^y^ (see column 2, Table 2), solving the linear problem [28]:$$\left(\begin{array}{ccc} N_{Ag}^{Aca} & 0 & N_{S}^{Sul}\\ 0 & N_{Bi}^{BiS} & N_{S}^{Sul}\\ 0 & 0 & N_{S}^{Sul} \end{array}\right)\left(\begin{array}{c} \mu_{Ag}\\ \mu_{Bi}\\ \mu_{S} \end{array}\right)=\;\left(\begin{array}{c} E_{Aca}\\ E_{BiS}\\ E_{Sul} \end{array}\right).$$

This means that the chemical potentials *μ*_α_ are calculated in a condition in which acanthite, Bi_2_S_3_, and S are at equilibrium.

The MD simulations of the various materials, necessary for the simulation of the XAFS spectra, were carried out in the framework of the DFT method, as implemented in VASP. The starting points were the relaxed cells obtained previously. The runs lasted about 10 ps (longer than the typical optical phonon periods, expected to be <1 ps) and were carried out in steps of 1 fs. Although no structural evolution was expected in the MD runs, we have verified that the total energy in the last portion of the run was stabilized and that the temperature distribution in the frames peaked at the desired value. The temperature was stabilized using a Nosé thermostat set at 300 K and monitored along the full run.

The structures from the last 2.9 ps of the run were used for the simulation of EXAFS spectra at the Ag K-edge [29]. In particular, from 290 single structures (1 taken every 10 fs), a single EXAFS spectrum was calculated with the FEFF 8.4 code [30], each Ag atom in the cell contributing with its own spectrum. Single and multiple scattering paths were considered up to a maximum path length of 10 Å and a maximum scattering order of 4, and the data range in k space was 0–20 Å^−1^. The data were then averaged and the final residual (i.e., the difference between the spectra taken with 289 and 290 frames) was about 3 × 10^−4^. The EXAFS spectra so obtained were successively filtered in the interval R = [0.4–6.8] Å in order to eliminate signals from atomic replicas generated by the periodic boundary conditions.

## 3. Results and Discussion

The background-subtracted Ag K-edge EXAFS spectra of all samples and of Ag_2_S are shown in Figure 3: They are all very similar and resemble the Ag_2_S spectrum. The main difference is the absence of a shoulder at ~5.1 Å^−1^ in the samples (indicated by the arrow); presumably, this is due to the presence of Bi in the local coordination around Ag. The Fourier Transforms (FTs) of the EXAFS spectra, performed in the range of 2.0–10.5 Å^−1^ of Ag_2_S and sample 2, which are also quite similar, are shown in Figure 4 as black continuous lines. The XANES spectra at the Ag K-edge is somewhat featureless because of the core–hole broadening. The experimental spectra of the samples and Ag_2_S are reported in the Appendix A (Appendix A) and also show similarities. We also report (Appendix A) a comparison between the experimental spectrum and FDMNES simulations for the schapbachite (cubic), acanthite (monoclinic), and matildite (hexagonal) bulk phases. There are similarities of the experimental spectrum with the hexagonal and monoclinic phases, but it is not possible to draw firm conclusions from these simulations, also because of the core–hole broadening.

The Ag_2_S spectrum was fitted using the monoclinic structure described above. As mentioned, Ag has two structurally inequivalent sites and, especially for site II, there is a rather broad distribution of interatomic distances around Ag. This fact complicates the data analysis. We, therefore, adopted a simplified approach, which, despite being an approximation, is able to reproduce well all spectra. Specifically, we fitted the spectra with the local structure of only the Ag–I site as the absorbing atom; the rationale is that the average of all scattering paths giving rise to the EXAFS signal relative to both Ag sites is equivalent to the sum of scattering paths for the more ordered Ag–I site. Clearly, we do not claim that the distances and Debye−Waller factors obtained in this way reproduce faithfully the full atomic distribution around Ag; however, the adopted approach is able to reproduce key features and trends in the local structural parameters and has the advantage of simplicity.

EXAFS scattering paths were calculated ab initio using FEFF, and the highest amplitude ones were included in the fit, which was performed in the ranges k = 2–10.5 Å^−1^ and R = 1–3.5 Å. The interatomic distances, an energy origin shift, and Debye−Waller factors were considered as fitting parameters; the many body amplitude reduction factor was fixed to the value obtained from a fit of a Ag metal reference spectrum (*S*_0_^2^
*= 0.834*). We found that the Ag_2_S spectrum could be fitted with single scattering paths due to two S atoms at ~2.50 Å (Ag–S1), one S atom at ~3.06 Å (Ag–S2), seven Ag atoms at ~2.97 Å (Ag–Ag1), and two more Ag atoms at ~3.11 Å (Ag–Ag2). These contributions give rise to the first two peaks in the FT of the spectra reported in Figure 4A, in which the components are labelled as just described. There are nine Ag atoms at different distances in the cation coordination shell, and we found that slightly different groupings of nine Ag atoms (for example, six plus three instead of seven plus two) did not give any significant differences except small variations in the Debye−Waller factor; clearly, in this approach, we are approximating the real distribution with a sum of two Gaussian components with different coordination numbers. The best fit and the individual components are shown in Figure 4A.

A model derived from the monoclinic structure of Ag_2_S was considered to fit the EXAFS spectra of the samples. As shown in Figure 2 (right panel), the Ag–II atom was replaced with Bi, and the corresponding theoretical scattering paths were generated and used in the fit. The spectra of the samples were fitted as for Ag_2_S with a modification regarding the Ag–Ag contributions. The cation contribution was thus considered as a linear combination of Ag–Ag and Ag–Bi scattering paths with coordination numbers 9*y* and 9(1 *− y*), respectively, and 0 ≤ *y* ≤ 1; in this way, the total coordination of cations around the central Ag atom in the alloy is 9, as in AgS_2_. We define normalized Ag–Ag and Ag–Bi coordination numbers as N_Ag−Ag_ = *y* and N_Ag−Bi_ = *1 − y*, respectively. The best fit for sample 2 is shown in Figure 4B, including the individual components (note that Ag–Bi substitutes Ag–Ag2) and the fits for all samples, for all of which the R-factor was less than 0.01 are shown in Appendix A.

The most important numerical results of the fit are reported in Table 3. We note that the Ag–S bond length is in the range of 2.48–2.51 Å, which is compatible with a low S coordination number in the first shell as in monoclinic acanthite and not with a high coordination site as in cubic schapbachite or hexagonal matildite [31]. This is a further confirmation that the local structure has features similar to those of the monoclinic phase. Furthermore, as shown in the Appendix A, the Bi L_III_ edge EXAFS spectra were fitted using the same model, albeit only in the first shell, and the results are reasonably compatible with the Ag K-edge results. Appendix A reports the FT and fit for a Bi_2_S_3_ reference sample and Appendix A the FTs and fits for the samples.

In Figure 5, we report the normalized Ag–Ag and Ag–Bi coordination numbers, as defined above, with the Ag concentration *x*. It is evident that N_Ag−Ag_ is always greater than *x* and N_Ag−Bi_ is always smaller than 1 *− x*, indicating that there is a tendency for cation ordering (i.e., clustering of similar cations). In order to quantify the degree of cation ordering, we calculated the Cowley [32,33] short-range order parameter, defined as
$$\alpha =1-\frac{y}{x}.$$

The value of *α* can vary in the interval between −1 and 1; *α* = 0 indicates a random distribution (*y* = *x*), negative values indicate an excess of Ag–Ag correlations with respect to the random case (like-atom clustering), and positive values indicate cation ordering (preference for Ag–Bi correlations); values of *α* are reported in Table 3. The value of *α* for all samples indicates a tendency for like-atom clustering, suggesting the presence of Ag-rich and Bi-rich volumes within the NPs.

The indication from XAFS that the local structure is similar to that of a monoclinic phase raises a serious issue as the XRD patterns exhibit peaks that, despite considerable broadening, are close to those of matildite and schapbachite and have no correspondence to those of monoclinic acanthite. In fact, in Figure 6, we report the experimental XRD patterns for sample 2 (the others are very similar) and simulations for the three structures. In all cases, the domain shape is assumed to be spherical, with a lognormal distribution of diameters (D), and with a small but non-negligible microstrain component of line broadening. Here, and in the following modelling, the background was represented by a Chebyshev polynomial of degree 12 to account for incoherent scattering, noise, and a fraction of non-crystalline phase. The acanthite pattern is so far from the experimental one that it cannot be refined (in the simulated pattern of Figure 6A, mean domain size was set to <D> = 5 nm, with a distribution standard deviation (s.d.) of 1.5 nm). Instead, the schapbachite and matildite simulations are close to the experiment pattern. Refined mean domain size <D> is 2.9 and 3.1 nm, respectively, for the cubic and hexagonal phases, with s.d. = 1 nm in both.

An apparent serious contradiction is, therefore, present. XAFS indicates a local structure around Ag cations similar to monoclinic acanthite; the detection of short (~2.48 Å) Ag–S bonds is particularly significant as EXAFS is a reliable method to measure interatomic distances in the first shell. On the other hand, the positions of XRD diffraction peaks are also, undeniably, a reliable fingerprint of the presence of particular crystallographic phases, even for NPs characterized by large particle-size-induced peak broadening and strain.

Ab initio DFT and MD simulations provide original insight to solve this contradiction. Analysis of the DFT-relaxed structures of the model compounds (acanthite, Bi_2_S_3_, and S) revealed an accuracy in the determination of the (Ag, Bi)-S distances of 1% with respect to experimental values and will not be discussed here. Table 4 reports some data retrieved from the static DFT (formation energy) and DFT−MD (structure) modelling of candidate structures to be considered for the interpretation of the XAFS data. An image of the structure resulting from the DFT−MD simulations is reported in Appendix A. The results of the formation energy confirm the observation of Viñes et al. [6] that matildite is more stable than schapbachite. However, these values should be considered just from a qualitative point of view as they are relative to a 0 K calculation and to a specific equilibrium configuration. Due to the complex environments around the cations involving many sites, structural data were derived from the first peak of the partial radial pair distribution functions (PRPDFs) obtained from the MD runs. In particular, the first peaks in the Ag−S and Ag−Ag PRPDFs were fitted with two Gaussians and the results are presented in Table 4.

It must be noted that in the random matildite structure, “short” Ag–S bonds appear. This is illustrated in Figure 7 in which the PRPDFs of matildite and random matildite are compared. EXAFS spectra were calculated based on the MD simulations, as described in the previous section; in Figure 8, we report the comparison between the experimental spectrum of sample 2 and the simulations for the four candidate structures: Bi-substituted acanthite (AcB), matildite (Mat), matildite with random cation occupation (MaR), and schapbachite (Sch). It is clear that the only candidate structure for which there is a good comparison in terms of overall lineshape is the random matildite one. As a final check, atomic coordinates from the DFT–MD simulations were used to simulate the XRD patterns, and the result is reported in Figure 9. Here, the hexagonal structure of Ag_12_Bi_12_S_24_ random matildite was mapped in the P1 space group, and the lattice parameters of the 2a, 2b, c supercell were refined starting from values for the standard hexagonal phase (a = b ≠ c, *α* = *β* = 90°, *γ* = 120°). The best fit shows a visible improvement with respect to the standard matildite of Figure 6C, with the statistical quality index Rwp decreasing from 3.05 to 2.55, and a corresponding improvement in the goodness of fit (GoF) from 2.01 to 1.72 [34]. Unit cell parameters change slightly from the starting hexagonal values to: a = 0.8119(5) nm, b = 0.8127(5) nm, c = 1.9735(10) nm, *α* = 91.0(1)°, *β* = 90.2(1)°, and *γ* = 119.1(2)°. The domain size is <D> = 3.1(3) nm, with distribution s.d. = 1.1(2) nm, values not far from the modelling of Figure 6 and TEM observations (cfr. Figure 1). The inhomogeneous strain is small but measurable (~0.0003), and is likely due to the surface relaxation commonly observed in nanocrystals [22,35]. The fact that random matildite is the preferential structure in the synthesized AgBiS_2_ nanocrystals, as revealed by our results, may have consequences in key properties relevant to the photovoltaic performance of this material, such as the electronic structure and the energy band gap. Revision of previously reported hybrid DFT calculations (which were done with the pure matildite structure) [6] would bring further insight to evaluate the impact of the found distortions. Although dramatic changes are not expected, such variations could explain the moderate mismatch between theoretical and experimental data.

Summarizing, DFT–MD simulations show that a random distribution of Ag and Bi in the cation sites of the hexagonal matildite structure gives rise to “short” Ag–S bonds detected experimentally by EXAFS; at the same time, the lattice remains hexagonal, providing a good comparison to the XRD data. The EXAFS analysis indicated that the distribution of cations around Ag is not actually random, but a preference for Ag–Ag correlations is present. However, more than 10% Bi is always found in the local cation environment of Ag, enough to induce strong local distortions in the Ag–S, producing short bond lengths that are not present in the bulk matildite structure. It is possible that these local distortions are related to electronic defects that affect the photovoltaic properties, a subject worthy of further investigation. Finally, these results illustrate the power of combined experimental–simulation studies of complex structures found in nanostructured materials.

## Figures and Tables

**Figure 1 nanomaterials-10-00316-f001:**
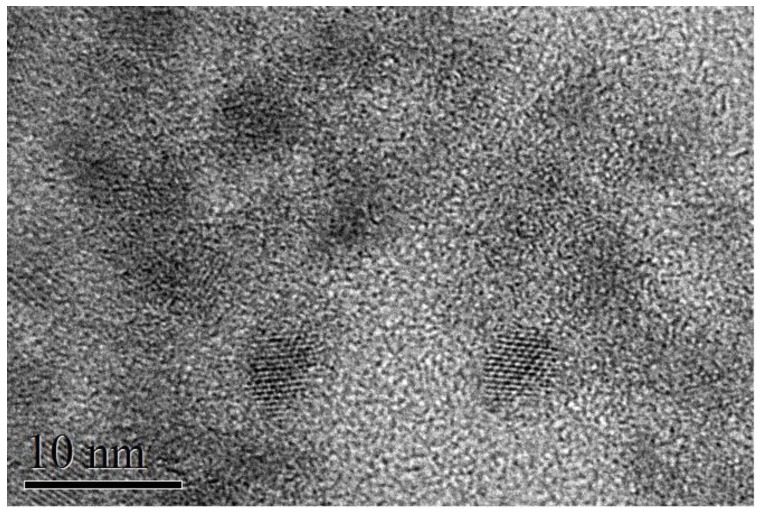
High-resolution transmission electron microscopy image of typical Ag_x_Bi_1−x_S_2_ nanoparticle (NP) thin films.

**Figure 2 nanomaterials-10-00316-f002:**
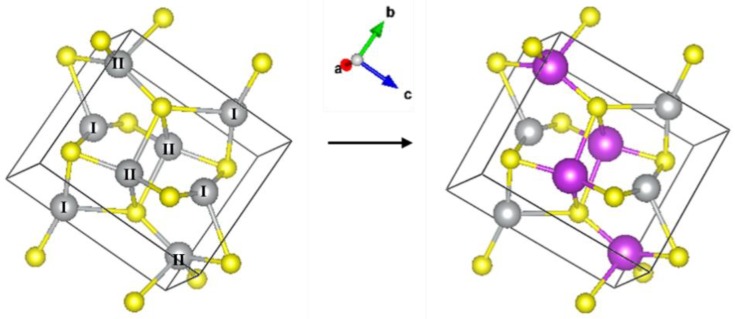
(**Left**): Monoclinic structure of Ag_2_S (● Ag ● S); the two inequivalent Ag sites are labelled I and II. (**Right**): Substitution of the second Ag site with ● Bi.

**Figure 3 nanomaterials-10-00316-f003:**
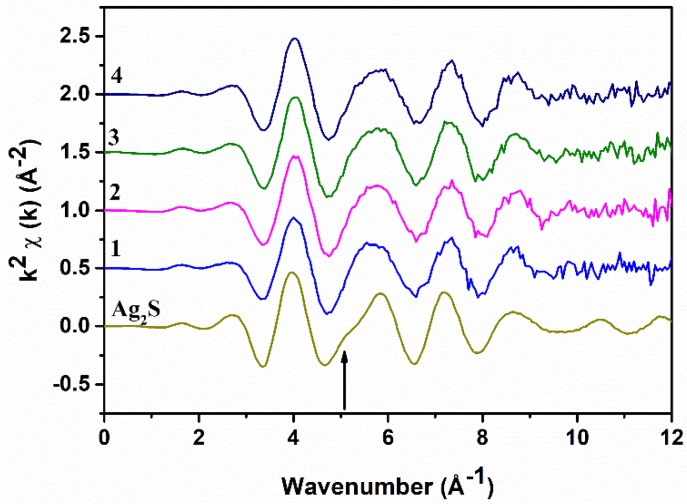
Ag K-edge EXAFS spectra of all AgBiS_2_ samples and Ag_2_S.

**Figure 4 nanomaterials-10-00316-f004:**
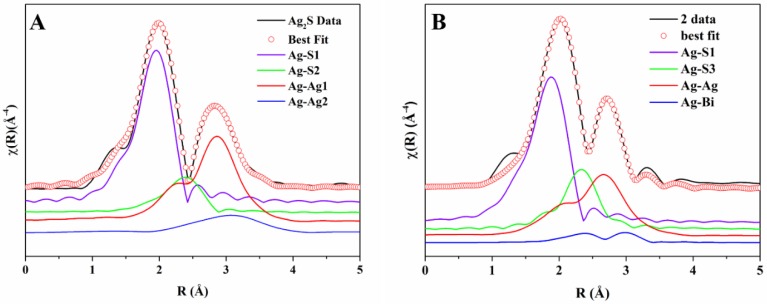
FTs of Ag K-edge spectra with components used in the fitting for Ag_2_S (**A**) and sample 2 (**B**). The labels are explained in the text.

**Figure 5 nanomaterials-10-00316-f005:**
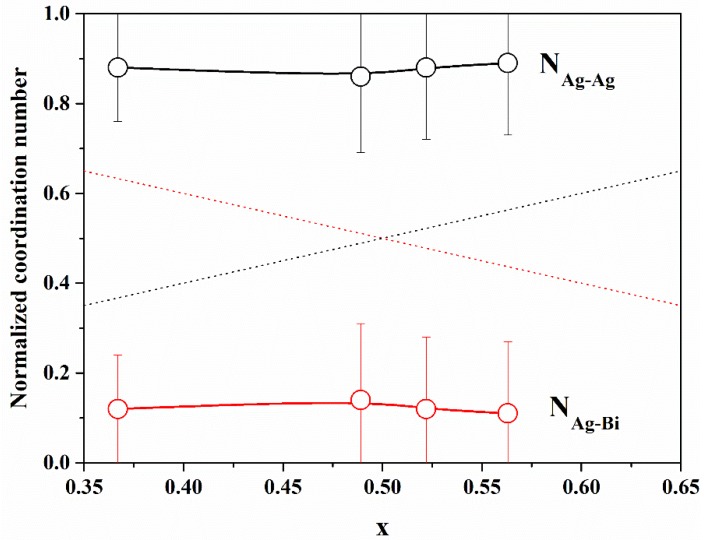
Normalized coordination numbers in the cation coordination shell as a function of Ag concentration *x*. The dashed lines correspond to a random distribution of cations, for which *y* = *x*.

**Figure 6 nanomaterials-10-00316-f006:**
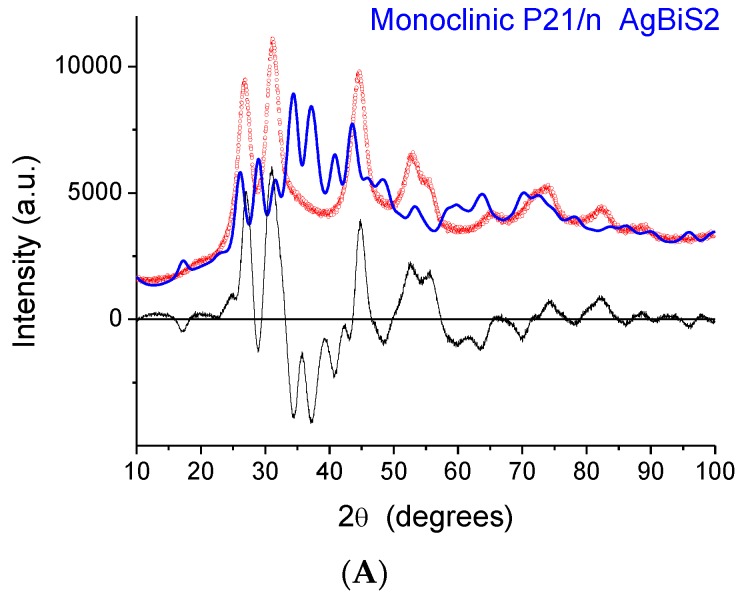
Comparison of the experimental XRD pattern for sample 2 (red points) with simulations (blue line) based on the acanthite (**A**), schapbachite (**B**), and matildite (**C**) structures. The difference between the experimental data and model (residual) is shown below (black line).

**Figure 7 nanomaterials-10-00316-f007:**
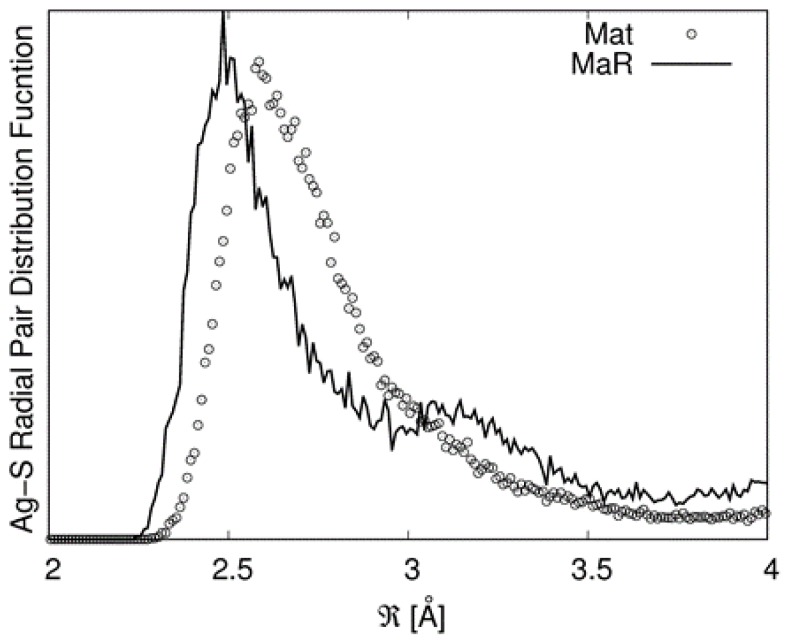
Ag–S PRPDFs for matildite (dots) and random matildite (black line).

**Figure 8 nanomaterials-10-00316-f008:**
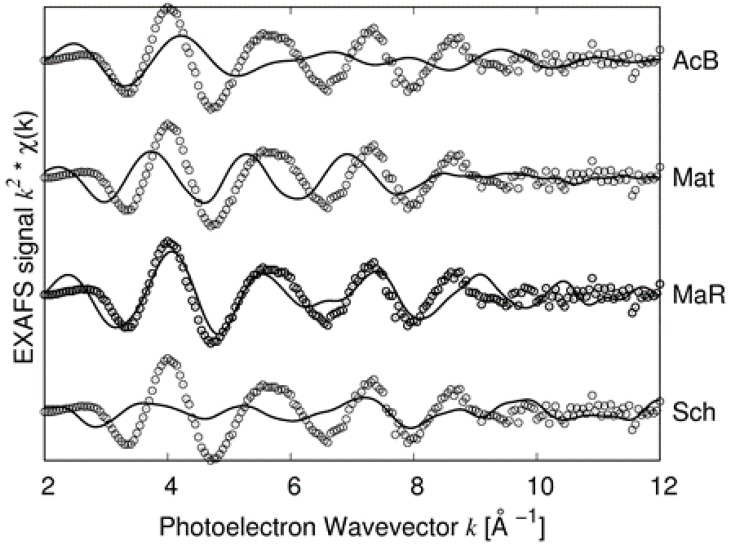
Comparison between experimental EXAFS spectrum of sample 2 (open circles) and simulated ones (continuous lines) based on four candidate structures: Bi-substituted acanthite (AcB), matildite (Mat), matildite with random cation occupation (MaR), and schapbachite (Sch).

**Figure 9 nanomaterials-10-00316-f009:**
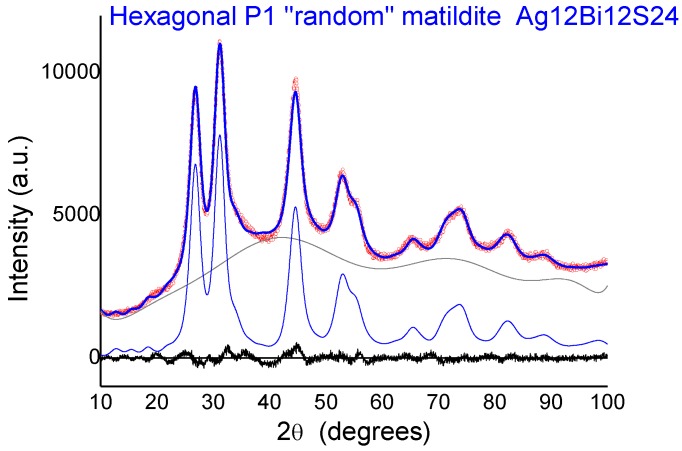
Comparison between the experimental XRD pattern of sample 2 (red points) and the simulation based on atomic coordinates obtained from the density functional theory (DFT)–MD simulations (blue line), including an amorphous-like background (grey). The difference between the experimental data and model (residual) is shown below (black line).

**Table 1 nanomaterials-10-00316-t001:** Ag concentration of the samples.

Sample Code	Ag Concentration (*x*)	
	Reaction Precursor	EDX Measured
1	0.38	0.37
2	0.50	0.49
3	0.55	0.52
4	0.59	0.56

**Table 2 nanomaterials-10-00316-t002:** Details of the cells used for the structural simulations. ICSD refers to the Inorganic Crystal Structure Database, and AMCSD refers to the American Mineralogist Crystal Structure Database.

Compound Code	Cell Composition	Total Number of Atoms	Starting Cell Dimensions (Å)	Structural Reference
AcB	40Ag 8Bi 32S	80	8.46 × 13.82 × 15.74	ICSD 44,507 [23]
Sch	16Ag 16Bi 32S	64	11.43 × 11.43 × 11.43	ICSD 604,842 [24]
Mat, MatR	12Ag 12Bi 24S	48	8.07 × 8.07 × 18.78	AMCSD 9218 [24]
Aca	8Ag 4S	12	4.23 × 6.91 × 7.87	Same as AcB
BiS	8Bi 12S	20	3.98 × 11.14 × 11.30	ICSD 201,066 [25]
Sul	32S	32	13.77 × 13.25 × 8.27	ICSD 43,251 [26]

**Table 3 nanomaterials-10-00316-t003:** Summary of quantitative analysis of Ag K-edge X-ray absorption fine structure in the extended energy range (EXAFS) spectra of AgBiS_2_ samples. Uncertainties in the least significant figures are reported in brackets (italics). *y* is the Ag concentration in the cation coordination shell and *α* is the Cowley short-range order parameter.

**Sample Code**	***R_Ag-S1_***	$\sigma_{Ag-S1}^{2}$	***R_Ag-S3_***	$\sigma_{Ag-S3}^{2}$	***R_Ag-Ag_***	$\sigma_{Ag-Ag}^{2}$	***R_Ag-Ag_***	$\sigma_{Ag-Ag}^{2}$	***R_Ag-Bi_***	$\sigma_{Ag-Bi}^{2}$	***y***	***α***
Ag_2_S	2.506 *(20)*	0.006 *(1)*	3.061 *(42)*	0.06 *(1)*	2.967 *(55)*	0.004 *(3)*	3.114 *(51)*	0.009 *(5)*				
1	2.481 *(13)*	0.010 *(1)*	2.914 *(26)*	0.004 *(2)*	2.874 *(38)*	0.029 *(4)*			3.175 *(49)*	0.011 *(8)*	0.85 *(17)*	−1.3 (0.3)
2	2.480 *(12)*	0.010 *(1)*	2.917 *(28)*	0.004 *(2)*	2.888 *(38)*	0.027 *(6)*			3.082 *(95)*	0.015 *(16)*	0.86 *(17)*	−0.8 (0.2)
3	2.473 *(13)*	0.009 *(1)*	2.910 *(24)*	0.006 *(3)*	2.908 *(38)*	0.028 *(5)*			3.22 *(10)*	0.015 *(17)*	0.88 *(16)*	−0.7 (0.2)
4	2.473 *(13)*	0.009 *(1)*	2.906 *(31)*	0.004 *(2)*	2.894 *(41)*	0.029 *(5)*			3.167 *(66)*	0.010 *(10)*	0.89 *(16)*	−0.6 (0.2)

**Table 4 nanomaterials-10-00316-t004:** Results from the ab initio structural modelling. Data from molecular dynamics (MD) runs are derived from the partial radial pair distribution functions (PRPDFs) where the first peaks were fitted with two Gaussians.

Compound	DFT Formation Energy Per Atom (eV)	MD Ag–S 1st Shell Distance ± HWHM (Å)	MD Ag–Ag 2nd Shell Distance ± HWHM (Å)
AcB	0.058	2.4 ± 0.1 2.6 ± 0.2	2.9 ± 0.1 3.4 ± 0.5
Sch	0.039	2.6 ± 0.2 3.0 ± 0.3	2.9 ± 0.1 3.2 ± 0.3
Mat	0.006	2.6 ± 0.1 2.9 ± 0.3	3.5 ± 0.4 4.3 ± 0.5
MaR	0.053	2.5 ± 0.1 3.0 ± 0.4	2.8 ± 0.1 3.0 ± 0.2

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
