# Peer review of "Cation Disorder and Local Structural Distortions in AgxBi1–xS2 Nanoparticles"

_nanomaterials, 2020, doi:10.3390/nano10020316_

Round 1
Reviewer 1 Report
Kesavan et al. study structure of Ag_xBi_1-xS_2 nanoparticles using x-ray and computer simulation methods. Their main result is that slightly atypical structure of nanoparticles can explain the seemingly controversial results from different experiments. This is a nice demonstration how computer simulations can be used to explain experiments, but the manuscript lacks discussion on impact or concequences of the results for the field. The studied nanoparticles are important for photovoltaics, but how the proposed structure would influence the relevant properties of the particles is not clear. Furthermore, description of some results, espacially in figures, do not fulfill the basic standards of scientific article, for example:
1) Figure 3: Line colors and labels are not explained.
2) Figure 4: Labels are not explained.
3) Several places in text refers to "sample 2", but this is not defined anywhere. This is a serious flaw which makes the evaluation of many results very difficult.
4) Figure 6: Caption refers to the "experimental XRD pattern for sample 2 (red)", but the sample 2 is not defined (see above) and red line is not present in figure.
5) Figure 8: Lines and points are not defined, labels on right are not defined.
6) Figure 9: Lines are not explained.
Finally,
7) The manuscript proposes a new type of lattice structure (main and only result), but figure of the structure is not shown. I think this is a must in this kind of manuscript.
Reviewer 2 Report
The authors employ simulations to study the cation disorder in the structure of AgxBi1-xS2 nanoparticles and explain a contradiction between the evidence provided by x-ray absorption and diffraction measurements. They conclude that the strong local distortions is induced in the structure, which lead to short Ag – S bonds to cause the disorder in the cation sublattice. We believe that this work is helpful in the study of improving photovoltaic efficiency. Therefore, it is recommended to be accepted for publication in Nanomaterials.
Author Response
We thank this referee for the positive opinion on our work. There are no comments and therefore no responses.
Reviewer 3 Report
This manuscript by Kesavan and colleagues reports on experimental and computational investigation of Ag-Bi-S compounds with a view to rationalising structural features of potential photovoltaic materials. The work will probably be publishable after modification.
The introduction talks about AgBiS2 a lot. It then jumps to looking at compounds with different compositions, which is what the bulk of the paper is about. No link is established between them. Why would one want to study these different cation ratio compounds? I assume that it is in order to draw insight into possible local structure within composition-modulations in AgBiS2. That in itself is problematic, as there is an assumption that such modulations do not occur in the non 1:1 compounds. But that argument is never made anyway. The link between AgBiS2 and the current work must be clearly articulated for this work to have a point.
It is stated that "The correspondence between sample code and Ag concentration is summarized in Tab. (sic) 1." This is not true. Table 1 only shows the resulting ratios, and assigns a serial number to the samples. The relationship between the reactant concentrations and the resulting compound should be given. I am also surprised at the cleanliness of the ratios. How were the atomic ratios in the synthesised samples confirmed?
There are several issues with how the DFT calculations were set up. Firstly, the Ag2S structure has two inequivalent Ag sites. Thus there are six possibilities for the substitutions and deletions to form AcB simply in terms of Ag I/Ag II selections, which gets multiplied considerably when symmetry breaking is taken into account. Which configuration was used? Why? Why were the others apparently not considered? This must be addressed in the manuscript, for this compound and others. The actual configurations used should be illustrated in the SI.
Secondly, the k-point convergence criterion is flawed. Comparable formation energy calculations require absolute convergence of the energies, as cancellation of error cannot be relied upon. The criterion of a change smaller than 1/1000th of the total energy is not an equivalent criterion for the different phases that differ by more than a factor of 6 in the number of atoms (close to an order of magnitude), even if one considers all elements to be equal (which they are not). This inequivalent convergence probably doesn't invalidate the results, but as a rationale it is erroneous.
Thirdly, how was the sufficiency of the length of the MD calculations tested? It is insufficient to, for example, argue that these timescales have been used in other publications. Convergence of the resulting quantities, as well as satisfactory equilibration, must be demonstrated explicitly.
There is no Figure 5B (line 214). In Figure 5, what are NAg-Ag etc.? These don't seem to be defined anywhere. Nor is what is meant by normalised coordination numbers.
